# The Impact of Migration Experience on Rural Residents’ Mental Health: Evidence from Rural China

**DOI:** 10.3390/ijerph20032213

**Published:** 2023-01-26

**Authors:** Li Deng, Xiaohua Hou, Haiyang Lu, Xuefeng Li

**Affiliations:** 1School of Finance, Sichuan Vocational College of Finance and Economics, Chengdu 610074, China; 2Institute of Western China Economic Research, Southwestern University of Finance and Economics, Chengdu 610074, China

**Keywords:** migration experience, mental health, returning migrant workers, China

## Abstract

Migration experience is considered to be an important factor affecting mental health. With the increasing number of rural-to-urban migrant workers returning to their hometowns, the impact of migration experience on rural residents is worthy of noting. Using the data from the 2018 China Labor Dynamics Survey, this paper took migration experience as the identification criteria for returning migrant workers and empirically examined the impact of migration experience on rural residents’ mental health. Our results indicated that migration experience had a significant negative impact on the mental health of rural residents. That is, returning migrant workers had a worse mental health status than that of rural residents who never left their hometowns. Mechanism analysis showed that social support and social comparison played an intermediary role in the impact of migration experience on the mental health of rural residents. We also detected considerable heterogeneity in the effects of migration experience: the short-term returning migrant workers and the passive returning migrant workers are more likely to be negatively affected by the migration experience. Our results emphasized the mental health problem faced by returning migrant workers. The policy makers should strengthen psychological education and mental health consultation according to the intergenerational differences and individual characteristics of returning migrant workers.

## 1. Introduction

Since the reform and opening-up policy in 1978, China has undergone a rapid urbanization process, with large numbers of rural workers migrating from rural areas to urban areas. As documented by Su et al. [1], China’s urban population has dramatically increased by more than 500 million people since 1979, among them, more than 75% of the increase is due to the migration from rural areas to urban areas. However, migrant workers, as major contributors to urban economic development, cannot enjoy the same welfare as urban residents due to the difference in the household registration system. Studies have found that migrant workers cannot get the same return as urban residents in the same job [2]. What is more, migrant workers lack social communication channels, civic entitlements, face discrimination from some locals and difficulty in truly integrating into society [3]. In the process of working in urban areas, migrant workers receive more income than those working in rural areas, but also show obvious health loss. The deterioration of health conditions leads to a significant reduction in working hours and a significant decline in the rate of return per unit of time [4]. Therefore, a large number of scholars have called for attention to the health problems of migrant workers.

Mental health is an important component of human capital [5,6]. Mental illness can trap individuals into disadvantageous positions in labor and marriage markets [7], and impose high costs on society [8]. In China, the mental health of migrant workers has gradually attracted extensive attention [9,10]. Most studies believed that Chinese migrant workers bear the pressure of work, family, and interpersonal relationships, so they have a high incidence of mental illness [11]. For instance, Hoi et al. found that the lack of a sense of belonging to the city, deferred pay, and working against one’s will under high-pressure management all increased the risk of mental illness of migrant workers [12]. Similar conclusions are also confirmed by other scholars [13,14]. Jie and Nikolas found that social exclusion was negatively associated with migrants’ mental health; the most important factors were the limited availability of full employment rights and the experience of stigmatization, discrimination, and injustice in the society [15]. Therefore, improving the mental health status of migrant workers is becoming an essential issue in China.

Returning migrant workers are an important component of migrant workers, which refers to the rural residents who return to rural hometowns after working in cities for a certain period of time. In China, the phenomenon of migrant workers returning to their rural hometowns from the city has always existed, and it has become more and more obvious in recent years. According to a 2009 survey by the National Bureau of Statistics of China, 55.14% of migrant workers have the desire to develop and settle in cities, but only 10.8% of migrant workers (14.3 million) have the economic ability to become urban citizens, which means that most of them are only living temporarily in the city and have to return to their hometowns in the future [16,17]. It has been found that returning migrant workers have played a positive role in improving the output efficiency of agricultural land, promoting entrepreneurial activities in rural areas, and strengthening the supply of professional services [18,19]. However, few studies have focused on the “migration sequelae” of returning migrant workers, especially for mental illness. Considering the persistence of mental illness, it is of great significance to study the mental health of returning migrant workers to improve the overall welfare of rural residents.

In order to confirm whether there are “migration sequelae” in the mental health of returning migrant workers, this paper compared the mental health status of returning migrant workers with that of other rural residents. Specifically, using the data from the 2018 China Labor Dynamics Survey (CLDS), this paper took migration experience as the identification criteria for returning migrant workers, and empirically examined the impact of migration experience on rural residents’ mental health. The marginal contribution of this paper is mainly reflected in the following three aspects: (1) Although some studies explored the mental health of migrant workers in an urban area, few studies focused on the returning migrant workers. This paper provides a nationally representative analysis of the impact of migration experience on mental health for rural residents, which is helpful for us to understand the mental health status of returning migrant workers. (2) This paper examines the mechanism of the impact of migration experience on the mental health of rural residents from the two aspects of social support and social comparison. The conclusions of the mechanism test is not only to provide guidance for improving the mental health of returning migrant workers, but also to provide useful information for understanding the challenges faced by migrant workers after returning home. (3) In this paper, returning migrant workers are divided into different types, including voluntary return, passive return, long-term return, and short-term return. To explore the mental health status of different types of returning migrant workers, respectively, is helpful for policy makers to formulate accurate assistance measures.

The next section reviews the relevant literature. Section 3 describes the data and the methods we employ in empirical analysis. In Section 4, we present our empirical results. Section 5 discusses our key findings, and Section 6 concludes.

## 2. Literature Review

### 2.1. Measure of Mental Health

There is no unified measurement standard for mental health at present. The commonly used methods include multi-indicator and single-indicator evaluation methods. As for single-indicator evaluation methods, psychological life satisfaction and psychological well-being are the widely used mental health indicators in the existing research [20]. The single-indicator evaluation methods may have shortcomings, such as low accuracy and one-sided measurement. Therefore, many scholars tend to adopt the multi-indicator evaluation methods. In general, the multi-indicator evaluation methods include CES-D (Center for Epidemiologic Studies Depression Scale) [21,22], BSI (Brief Symptom Inventory) [23], SCL-90 (Symptom Checklist 90) [24], and GHQ-12(General Health Questionnaire) [25], etc. These studies mainly use score summing or mean processing methods, factor analysis or hierarchical analysis, and other methods to deal with the self-evaluation indicators of mental health status and to finally calculate the mental health score [21,26].

Among these methods, CES-D is more suitable for Chinese rural residents, especially the returning migrant workers. In the process of China’s urbanization transformation, due to the extremely fierce competition, the fast pace of life and work, and high pressure, depression has become an important disease in China beyond coronary heart disease. In addition to being a feature of high incidence, depression is also difficult to cure, is hyper recurrent, and highly disabling [27]. Many patients have suicidal ideation and suicidal behavior, which not only endangers the health and life of patients, but also brings a heavy burden to countless families and causes huge costs to society. The WHO (World Health Organization) predicts that depression will become the world’s top disease burden by around 2030 [28]. The CES-D has been developed by the National Institute for Mental Health (NIMH) in order to identify common depression symptoms in adolescents and adults by self-diagnosis. It includes a range of related symptoms that occurred over the past week. The CES-D has been demonstrated to be reliable and a valid screening tool for the symptoms of depression. The score of the scale was proven to be strongly correlated to clinical depression diagnoses [29]. In the 1990s, the CES-D was introduced to China and its validity was tested in a number of studies [30,31].

### 2.2. Rural Residents’ Mental Health

According to the World Health Organization (WHO), mental health refers to a state of well-being in which the individual realizes his or her own abilities, can cope with the normal stresses of life, can work productively and fruitfully, and is able to make a contribution to his or her community [32]. The current survey and research results on the mental health status of rural residents in China show that, compared with urban residents, rural residents are limited by economic conditions, backward infrastructure, lagging psychological services and other factors, and their mental health status is generally poor, especially the mental problems of anxiety and depression are very prominent [33,34]. For example, the Report on China’s National Mental Health Development (2020) released by the Institute of Psychology of the Chinese Academy of Sciences shows that there is a significant difference between the mental health status of rural residents and urban residents, and the detection rate of high risk of depression among rural residents is higher than that of urban residents. Some rural residents have obvious irritability, irascibility, indifference, anxiety, jealousy, fear of competition, fear of risk, depression, and other negative emotions. Zhang et al. have noticed that rural residents in various regions of China have a higher incidence of mental problems than urban residents, and depression is more prevalent and prominent [35]. He et al. showed that the mental health of older rural adults had become a major public health concern in China [33].

Existing literature on the factors affecting the mental health of rural residents mainly focuses on individual family characteristics and social environmental factors. Among them, in terms of individual characteristics, research shows that gender, education status, age and marital status, and other demographic factors significantly impact the mental health of rural residents [36,37,38]. In terms of social environmental factors, occupation, financial situation, and social network are the main factors affecting the mental health of rural residents [39,40]. At present, social environmental factors are increasingly being studied by scholars. Mental health is increasingly seen as a social issue, with social factors even more influential than factors such as medical technology [41]. For example, Kawachi et al. [42] stated that low levels of social trust led to low levels of mental health. A weak attachment among members of a group may result in a lack of support relationships, which may lead to worse mental health status [43]. However, the existing literature still ignores a key factor affecting the mental health of rural residents: the migration experience.

Migration is a stressful process with many changes. Migration brings not only physiological changes, but also social and cultural changes [44]. Previous studies have almost agreed that migration experience is a critical factor affecting the mental health of rural residents who migrate from rural to urban areas [45]. For instance, Fennelly found that migrant workers had better mental health outcomes than the native-born workers when they first arrived in their new countries of residence; however, with increasing time in the host society, their mental health advantage diminished significantly [46]. Lam and Johnston used CES-D and found migrants were more likely to have clinically significant depressive symptoms than urban residents [22]. Li et al. examined the mental health symptoms of rural-to-urban migrants in Beijing, and reported that rural-to-urban migrants suffered from poorer mental health status than their counterparts in the rural areas to which they immigrated [11]. Some studies have further explored the mechanism of the impact of rural-to-urban migration experience on the mental health of rural residents. Yue and Wang adopted the Survey Data of Migrant Workers (2016) and pointed out the social support and social comparison had an important impact on the mental health of rural-to-urban migrant workers [47]. Chen et al. noticed that rural-to-urban migrants experienced stress from frequent overtime work, poor working conditions, the lack of social security, interpersonal-related difficulties, more social pressure, and exclusion, and thus showed a high incidence of mental problems [48].

These studies provide evidence for understanding the current situation and determinants of rural residents’ mental health. However, existing literature only focuses on the impact of migration experience on the mental health of rural-to-urban rural residents, but ignores the returning migrant workers, who experienced the two-way migration of “rural-to-urban” and “urban-to-rural”. With the increasing proportion of migrant workers who have returned to the countryside and become rural residents again due to industrial transformation and upgrading, rising labor costs, and other reasons in recent years, the impact of the urban-to-rural migration experience on rural residents is also very worthy of noting.

## 3. Materials and Methods

### 3.1. Data

The data for our analysis come from the 2018 Chinese Labor Dynamics Survey (CLDS) conducted by the Social Science Survey Center of Zhongshan University. The CLDS is an influential, representative, and widely used micro-survey database. The survey objects are the labor force aged 15–64 in sample households. Probability sampling methods were used at multiple stages and levels proportional to the size of the labor force to determine the sample. A total of 16,537 labor force individuals from 13,501 households in 368 communities of 29 provinces (except Hong Kong, Macao, Taiwan, Xizang, and Hainan) completed the survey. In order to explore the mental health problems of returning migrant workers, we focus on the impact of the urban-to-rural migration experience on the mental health of rural residents. The samples used include those living in rural areas and with a rural hukou. The observation of missing values was removed from the sample. As a result, 7790 observations were used for analysis after sample selection and data cleaning. The sample size of regression analysis varies according to different specifications of the model and depends on the data availability of the variables involved.

### 3.2. Measures of Mental Health

The dependent variable was the mental health of rural residents measured by the 20-item CES-D scale. In CLDS’s questionnaire, respondents were asked, “how often have they felt the following emotions in the past week?” (see Table A1). In this paper, referring to Lu and Chen et al. [21,48], the mental health status of rural residents was first measured by an orderly classification variable (MH) based on the CES-D, using the multi-indicator evaluation method. The response categories ranged from 0–3, including “Almost none” (0), “Rarely” (1), “often” (2), and “all the time” (3). The sum score from CES-D ranges from 0 to 60, with a higher score indicating more depressive mental health.

As a robustness check, we further measured the mental health of rural residents by a dummy variable. As long as the respondents chose “all the time” in one of the 20 items, the dummy variable the mental health of rural residents equaled 1; otherwise, it equaled 0. Alternatively, as long as the respondents choose “all the time” or “often” in one of the 20 items, the dummy variable equaled 1; otherwise, it was 0. In addition, as a single indicator measurement, psychological life satisfaction and life happiness are frequently used in existing studies. Following previous studies [20], this paper also adopted these two single indicators for the robustness test.

### 3.3. Measures of Migration Experience

In CLDS’s questionnaire, respondents were asked, “do you have migrant experience (cross-county migration for more than half a year)”. Based on this item, a dummy variable Migration experience (ME) is constructed. If the respondent has migration experience, the value is “1”; otherwise, it refers to the rural residents who have never gone out, and the value is “0”. As mentioned above, returning migrant workers refer to the rural residents who return to rural hometowns after working in cities for a certain period of time. That is, rural residents with migration experience can be identified as returning migrant workers. Furthermore, if the migration experience has a negative impact on the mental health of rural residents, it means that the mental health of returning migrant workers is worse than that of other rural residents.

### 3.4. Other Covariates and Descriptive Statistics

The control variables are the potential factors that affect the mental health of rural residents. Specifically, following previous literature [47,49], we have two classes of control variables. One is individual family factors, and the other is social environment factors, including gender, age, education, marital status, religious belief, insurance status, exercise habit, the logarithm of annual house income, and the number of brothers and sisters. The descriptive statistics of the variables are presented in Table 1.

### 3.5. Empirical Strategies

#### 3.5.1. Baseline Model

To explore the nexus between migration experience and rural residents’ mental health, we estimate the following equation:*MH**_i_* = *α**_1_* + *β**_1_**ME**_i_* + *σ**_1_**Control* + *ξ**_i_*(1)
where *MH_i_* is the proxy for mental health status of rural resident *i*, and the core explanatory variable *ME_i_* is the indicator of whether rural resident *i* has had migration experience. *Control* is a vector of controls, and *ξ* is the error term. Clustered robust standard errors are used in this paper. Since OLS needs to assume a normal residual distribution, which is difficult to satisfy when the explained variables have ordered and discrete values, we first used an ordered probit (OP) model to estimate the equation, and we also used an ordered logit (OL) model as a robustness check when we used the alternative dependent variable.

#### 3.5.2. Intermediate Effect Test

Referring to prior work [47], this paper further used the following mediation effect model to perform the mechanism analysis of migration experience on the mental health of rural residents:*Channel**_i_* = *α**_2_* + *β**_2_*
*ME**_i_* + *δ**_2_**Control**_i_* + *ω**_i_*(2)
*MH_i_* = *α**_3_* + *β**_3_*
*ME_i_* +*μChannel_i_* + *δ_3_**Control_i_*+ *φ_i_*(3)
where *Channel_i_* is the mediation variable of our interest, including Social Comparison and Social Support. If the coefficient *β_2_* of *ME_i_* in Equation (2) and the coefficient *μ* of *Channel_i_* in Equation (3) are both significant, the mediating effect exists. α_2_ and α_3_ are the constant terms, *ω_i_* and *φ_i_* are the error terms, and *β_2_*, *β_3_*, *δ_2_*, *δ_3_*, *μ* are the parameters to be estimated.

## 4. Results

### 4.1. Effects of Migration Experience on Rural Residents’ Mental Health

Table 2 presents the results of the ordered probit (OP) model (1). Column 1 reports the results controlling for only gender and age. In columns 2 and 3, we further control for education, marriage status, religious belief, insurance status, and exercise habits. Column 4 reports the estimated effect of ME on MH, referring to all the control variables in Table 1. The results show that the coefficient of ME is positive and significant at the 1% statistical level from Column 1 to Column 4, indicating that the migration experience significantly deteriorates the rural residents’ mental health status. That is, the mental health of returning migrant workers is worse than that of other rural residents. Our main results remain unchanged after adjusting for those additional controls.

Regarding the control variables, we find that education, marital status, exercise habit, and household income have a negative effect on MH, which means that higher education, being married, higher household income, and doing more exercise can significantly improve the mental health status of rural residents. It is noteworthy that the effect of age on the mental health of rural residents shows an inverted U-shape.

### 4.2. Robustness Checks

To ensure the robustness of the baseline results, we conducted a series of robustness tests. Our first robustness check concerns the alternative measures of MH. In Table 3, we report the estimates of the ME on MH using a variety of proxies for the mental health of rural residents, other than MH, as the dependent variable. MH is replaced by two kinds of dummy variables (MH (a) and MH (b)) in the first two columns in Table 3. One is to assign a value of “1” to those who answer “most” and 0 to others in the 20 items (column 1). The other is to assign a value of “1” to those who answer “most” and “occasionally”, and 0 to others in the 20 items (column 2). The coefficient of ME is positive and significant at the 1% statistical level, which is consistent with the baseline results. Then, MH is replaced by single proxies of psychological life happiness (column 3) and psychological life satisfaction (column 4). In existing research, psychological life happiness and psychological life satisfaction are widely used as single measures of mental health [49]. The results demonstrate a negative impact of migration experience on the psychological life happiness and life satisfaction of rural residents, implying that migration experience deteriorates rural residents’ mental health. The results are consistent with the previous ones.

In Table 4, we further performed several tests to check if our results were robust to different specifications. In Panel A, we adjusted the clustering method to village-level clustering. Because the typical characteristic of Chinese rural palaces is still the acquaintance society, the behavior and intentions of different rural residents may be related to each other, and they may face some common interference factors. In this paper, it is assumed that the error items in the same countryside are correlated with each other, and in different villages are not correlated. Based on the baseline regression model, the cluster was adjusted to the village level. The results remained unchanged.

In Panel B, the robustness test was conducted by changing the distribution form of data, so we re-ran the regression by an ordered logit model. The results showed that the coefficient of ME was still positive and significant at the 1% statistical level after we changed the model set.

In Panel C, we used IV-ordered probit (IV-OP) to rule out the potential endogeneity of migration experience. Social networks and social relationships play a key role in individuals’ migration decisions [47]. The decision of whether rural residents go out to work or not, and whether they return to their hometowns to live is directly related to the community they live in. Referring to Démurger and Xu [50], we used the proportion of returning migrant workers in each village as a source of exogenous variation for the migration experience. The proportion of returning migrant workers in each village was calculated by the number of returning migrant workers in the community where the sample was located divided by the number of individual samples in the community. The instrument variable has significant explanatory power over migration experience, while it has no direct influence on rural residents’ mental health. The results documented the reliability of our findings.

However, there may be another concern. Because whether a rural resident has ever had the migration experience is not randomly assigned or exogenously decided, the returning migrant workers’ group may have individual characteristics. Meanwhile, the migrant workers who choose to return home are possibly those individuals with poor physical and mental health, so selection bias may exist. To alleviate this endogenous problem, we adopted the Heckman two-step method (Panel D). The selection equation examined the factors influencing the decision to return to the village. Referring to relevant studies [20], this paper selected the following variables as the exclusive constraint variables of the selection equation. These include gender, age, education, marital status, religious belief, insurance status, and exercise habits. These variables have a more direct effect on whether migrant workers return home or not. The results showed that the regression results of the Heckman two-step regression were consistent with the previous analysis. At the same time, the coefficient of the Inverse Mills Ratio was significant at 1% statistical level, indicating that the model effectively controlled the sample selection bias.

Finally, we used the propensity score matching method (PSM) to ensure that rural residents with and without migration experiences were comparable, which helped address the endogeneity issue of observed variables. Specifically, we used a one-to-one nearest neighbor matching of propensity scores without replacement, which was examined by a logit regression of the dummy variable on a set of control variables used in the main regression. After matching, we found no significant difference between rural residents with and without migration experiences. Then, we used the matched sample to examine the impact of migration experience on rural residents’ mental health. As shown in Panel E of Table 4, the estimated ATT is positive and statistically significant at 1% statistical level, which shows our results are robust.

### 4.3. Mechanism Analysis

According to prior literature, social comparison and social support are important factors affecting the mental health status of rural residents [44,47]. First, we conduct the mechanism analysis from the perspective of social comparison. Regression analysis was conducted according to Equations (2) and (3), using Comparison with relatives or Comparison with neighbors as intermediate variables, and the results are reported in Table 5. Respondents were asked in the questionnaire, “Do you think your current psychological well-being is better or worse than that of your relatives (or your neighbors)?” The response categories ranged from 1–5, including “Much lower” (1), “lower” (2), “almost same” (3), “higher” (4), and “much higher” (5), with a higher score indicating better psychological well-being compared with relatives or neighbors.

In Column 1 of Table 5, we examine the impact of ME on Comparison with relatives. The coefficient of ME is negative and significant at the 1% statistical level, indicating that compared with rural relatives without migration experiences, returning migrant workers believe their psychological well-being is significantly lower. The results in column 2 show that after adding the mediating variable Comparison with relatives by Equation (3), both the variables ME and Comparison with relatives have a significant impact on MH. Similarly, In Column 3 of Table 5, we examine the effect of ME on Comparison with neighbors. The coefficient of ME is also negative and significant at the 1% statistical level, indicating that compared with neighbors without migration experiences, returning migrant workers believe their psychological well-being is significantly lower. The results in column 4 show that after adding the mediating variable Comparison with neighbors by Equation (3), both the variables ME and Comparison with neighbors have a significant impact on MH. The results in Table 5 indicate that social comparison (Comparison with relatives or Comparison with neighbors) is an intermediary channel for the deterioration of rural residents’ mental health.

Then, we conducted the mechanism analysis from the perspective of social support. Regression analysis was conducted according to Equations (2) and (3), using Mutual assistance and Trust in neighbors as intermediate variables, and the results are reported in Table 6. As for Mutual assistance, the respondents were asked in the questionnaire, “Do you help your neighbors and other residents in your community (village)?” The response categories ranged from 1–5, including “Very little” (1), “little” (2), “general” (3), “relatively much” (4) and “very much “(5), with a higher score indicating rural residents felt more mutual support. As for Trust in neighbors, the respondents were asked in the questionnaire, “Do you trust your neighbors and other residents in your community (village)?” The response categories ranged from 1–5, including “Very little trust” (1), “somewhat little trust” (2), “general” (3), “somewhat trust” (4) and “very much trust “(5), with a higher score indicating a higher level of trust between neighbors.

In Column 1 of Table 6, we examine the impact of ME on Mutual assistance. The coefficient of ME is negative and significant at the 1% statistical level, indicating that compared with rural residents without migration experiences, returning migrant workers believe there is less mutual help between neighbors. The results in column 2 show that after adding the mediating variable Mutual assistance by Equation (3), both the variables migration experience and Mutual assistance have a significant impact on MH. Similarly, In Column 3 of Table 6, we examine the effect of ME on Trust in neighbors. The coefficient of ME is also negative and significant at the 1% statistical level, indicating that compared with rural residents without migration experiences, returning migrant workers believe there is less trust in the neighborhood. The results in column 4 show that after adding the mediating variable Comparison with neighbors by Equation (3), both the variables ME and Trust in neighbors have a significant impact on MH. The results in Table 6 indicate that social support plays a partial mediating role in the relationship between migration experience and rural residents’ mental health.

### 4.4. Heterogeneous Effects

The effect of migration experience on the mental health of rural residents may vary across different subpopulations. In Table 7, we explore whether the impact differs by return type. Different returning migrant workers may have various reasons and plans. Some may choose to live in rural areas for a long time, while some may continue to migrate, which leads to a different impact on mental health status. This paper, according to the question in the individual questionnaire “Do you still plan to go out for work”, identifies those who do not plan to go out for work in the future as long-term returning, and those who plan to go out for work in the future as short-term returning, and dummy variables Long-term returning and Short-term returning were, respectively, constructed. Then, we used Long-term returning and Short-term returning to replace ME in the model (1) and re-ran the regression. The results are reported in Table 7. Columns 1 to 3 of Table 7 show the coefficient is positive and significant at the 1% statistical level for both Long-term returning and Short-term returning, while the coefficient of the Short-term returning group is significantly larger, which indicates that the migration experience has a stronger impact on short-term returning migrant workers.

Then, we partition the sample into a voluntary returning group and a passive returning group. Prior research highlighted that the purpose of labor going out is to complete the accumulation of necessary economic and human capital. They can get better development opportunities after returning home, so they will choose to voluntarily return home after the completion of the established purpose. However, other scholars have suggested that only “losers” would decide to return home, because they were “eliminated” from the city due to old age, illness or other reasons, or were forced to return because they failed to find a suitable job in the city [51,52]. This paper, according to the individual questionnaire “Why do you plan to stay in your hometown for a long time”, further classifies the long-term returning migrant workers into the voluntary returning group or the passive returning group. Those who answered “to take care of my families”, “I can get a better income when I return to my hometown”, “I am used to local life but not used to living outside” were classified into the voluntary returning group, and those who answered “I cannot find a good job outside”, “I went home because of illness, injury, or other similar physical factors” were classified into the passive returning group, and dummy variables Voluntary returning and Passive returning were, respectively, constructed. Then, we used Voluntary returning and Passive returning to replace ME in the model (1) and re-ran the regression. The results were presented in Columns (4) to (6) of Table 7, and we found that the impact of migration experience is significant and stronger only for passive returning migrant workers; whereas, we found no significant impact for voluntary returning migrant workers.

In Table 8, we explore whether the impact of migration experience on the mental health of rural residents differs by age and village status. First, we classified the sample over 55 years old as Old aged, and others as Young and Middle-aged. Columns 1 and 2 of Table 8 show that the impact of migration experience is only significant for Young and Middle-aged rural residents; whereas, we find no significant effect for Old aged rural residents. Then, according to the “cleanliness of community appearance” in the community questionnaire, we identified the village samples with a score of “9–10” as Good Village, and others as Ordinary Village. Columns 3 and 4 of Table 8 show that the effect of migration experience is only pronounced for rural residents’ mental health in Ordinary Villages; whereas, we find no significant impact for those in Good Villages.

## 5. Discussion

The majority of the existing literature focuses on the mental health status of rural-to-urban migrants in cities, but few studies take into account the mental health of rural residents with migration experience. In this paper, we examine the effects of migration experience on rural residents’ mental health. The research of this paper enriches the literature on the determinants of rural residents’ mental health, which is helpful for us to understand the mental health status of returning migrant workers. It also has important policy implications for promoting the mental health of returning migrant workers who have made outstanding contributions to urban construction in developing countries and have effectively utilized the role of their accumulated human and physical capital.

Using the population micro survey data of the 2018 China Labor Dynamics Survey (CLDS), we find that migration experience significantly deteriorates the rural residents’ mental health status. The results in this paper are supported by Wang and Zhang, and Xu et al. [49,53], who showed that returning migrant workers had significantly lower levels of happiness. Rural-to-urban migrant workers experience stress arising from frequent overtime work, poor working conditions, lack of social security, interpersonal-related difficulties, more social pressure and exclusion, which lead to deteriorating mental health conditions when they are in cities [48]. Due to the persistence of their mental illness problems this vulnerable group of migrant workers is of greater concern and continuous concern when they return to their rural home; however, this may be helpful in improving the overall welfare of rural residents.

We used social support and social comparison mechanisms to analyze the possible reasons. The results showed that social support and social comparison played an intermediary role in the impact of migration experience on the mental health of rural residents. On the one hand, mental health is not only related to the income level, but also closely related to the selected social reference point in cognition [54]. These returning migrant workers have experienced a double migration from the countryside to the city and from the city back to the countryside; they are repeatedly surrounded by complex information, and they have to change their social points of reference in cognition repeatedly. Although returning migrant workers may have accumulated more wealth than left-behind residents, the reference point of most of them still stays behind the superior urban residents, resulting in the sense of relative deprivation and negative emotions, often manifested as anger, resentment, or dissatisfaction [55]. The migration experience has left a deep imprint in their hearts, which changes their evaluation of life status at the cognitive level and their subjective judgment of life at the emotional level. The combination of the relative deprivation caused by the horizontal and vertical way has a very negative impact on the mental health of returning migrant workers [56]. On the other hand, most returning migrant workers face the problem of reintegration. It usually takes a long time for them to readjust to working and living in the countryside [57]. Mechanism analysis shows that returning migrant workers believe their neighbors’ trust and support are significantly lower, indicating that returning migrant workers face difficulties in reintegration. The migration experience has distanced them from the remaining villagers and increased their sense of alienation. The weak attachment among group members may lead to low levels of supportive relationships; the lack of emotional support in social networks harms the mental health of returning migrant workers.

We also revealed that the effect was heterogeneous across different sub-populations. In particular, the impact of migration experience is more significant and stronger for short-term and passive returning migrant workers, whereas we find no significant impact for voluntary returning migrant workers. One possible explanation is that most of the voluntary returning migrant workers are those who completed their goals, or who returned home for family reasons. After returning home, they have more emotional support and better social support. In contrast, passive returning migrant workers and short-term returning migrant workers are more inclined to stay in the city. The passive returning ones had to return home because they contracted a serious health problem [58]. The social reference points in the cognition of these rural residents stay in the superior city group, which is not conducive to their mental health.

In terms of age and village status, we found that the impact of migration experience was only significant for young and middle-aged rural residents and for the ordinary village; whereas, we found no significant impact for old-aged rural residents and for the good village. As documented by Li and Liang [10], the mental health of the new generation with rural-to-urban migration experience is significantly worse than that of the older generation. With the improvement in education level, the new generation of rural residents have higher expectations for life, a deeper understanding of the social environment, increased spiritual pursuit, and thus a growing sense of loss after returning home, which might have a great impact on their mental health. Yang pointed out that [2], in addition to making money, the new-generation rural residents also had a strong desire to live like urbanites and to integrate into urban societies. In comparison, it is quite joyful to note that a good village moderates the mental health of returning migrant workers. A good village is not only outstanding in its appearance, but also outstanding in its spiritual civilization construction. Villagers help each other more frequently and neighbors are friendlier, which is conducive to the mental health of returning migrant workers.

Combined with the above analysis results, this paper constructs a characteristic analysis model of the influence of migration experience on the mental health of rural residents (as shown in Figure 1). Migration experience has a significant negative impact on the mental health of rural residents. That is, returning migrant workers have a worse mental health status than that of rural residents who never leave their hometowns. Social support and social comparison play an intermediary role in the impact of migration experience on mental health of rural residents. In addition, the short-term returning migrant workers and the passive returning migrant workers are more likely to be negatively affected by the migration experience. For young and middle-aged rural residents and rural residents living in villages with ordinary appearance, the negative impact of migration experience on mental health is more significant.

Several limitations of our work warrant discussion. First, this study is intrinsically a cross-sectional analysis and therefore fails to identify the long-term impact of migration experience on rural residents’ mental health. In the future, panel data can be used for empirical analysis if data collection is feasible. Second, mental health is a fairly complex concept. It has some limitations since we mainly used CES-D to measure it in this paper. Although we explained above that this measure is more applicable to rural residents in China, especially to the returning migrant workers, some useful outreach will be necessary in the future. Moreover, in the mechanism test section, we mainly used Social support and Social comparison as mediating variables. In fact, there are many mechanisms by which migration experience might affect the mental health of rural residents, and future studies will be needed to explore more possible impact channels.

## 6. Conclusions

Using the 2018 population micro survey data of the China Labor Dynamics Survey (CLDS), our findings confirmed the detrimental impact of migration experience on rural residents’ mental health, and these findings have some important policy implications. Mechanism tests showed that social support and social comparison played partial mediating roles. We also detected considerable heterogeneity in the effects of migration experience across returning type, age, and village status; the short-term returning group and passive returning migrant workers, as well as the young and middle-aged groups, and the ones who live in the ordinary village are more likely to be affected by the migration experience. The mental health problems of returning migrant workers with migration experience should receive more and broader attention.

At present, China is comprehensively promoting the strategy of healthy China and rural revitalization. A substantial number of migrant workers returning to their hometowns have alleviated the shortage of rural human capital and material capital. It is of great practical significance to pay attention to and improve the mental health condition of these rural residents with migration experience.

First, it is important to strengthen psychological education for returning migrant workers, and improve the counseling mechanism of their mental health. The cognitive reference point of most returning migrant workers still stays similar to that of more superior urban residents, thus generating a sense of relative deprivation, leading to the deterioration of mental health. Policy makers should increase capital input, gradually establish and improve rural mental health professional institutions and psychological rehabilitation institutions, recruit relevant talents to build professional psychological counseling teams, timely and scientifically provide psychological counseling and assistance services for returning migrant workers, and guide them to rationally view and evaluate the urban–rural gap.

Second, it is important to help returning migrant workers to establish a sound social support system. Most of the returning migrant workers face difficulties in reintegration. A favorable working and living environment for them is necessary. Policy makers should actively promote the employment and entrepreneurship of returning migrant workers, and guide their accumulated human capital to be effectively used in rural areas. At the same time, the mutual assistance mechanism between local villagers and returning migrant workers should be effectively established through the construction of village appearance and spiritual civilization, and to enhance the sense of belonging and value of returning migrant workers.

Third, it is important to pay attention to the differences between generations, and individual characteristics of returning migrant workers. Policy makers should emphasize the mental health problems of the new generation of rural residents returning home, and strengthen counseling and training for this group. They should guide them to rationally view and evaluate the gap between urban and rural areas, and give full play to the human capital and material capital advantages of young and middle-aged returning migrant workers.

## Figures and Tables

**Figure 1 ijerph-20-02213-f001:**
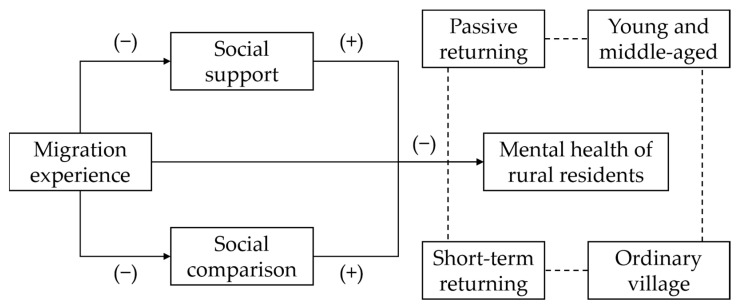
The impact of migration experience on rural residents’ mental health.

**Table 1 ijerph-20-02213-t001:** Descriptive statistics.

Variables	Mean	Std. Dev.	Min	Max
Mental health (the sum of the 20 items)	8.025	9.452	0.000	60.000
Migration experience (1 = yes; 0 = otherwise)	0.213	0.410	0.000	1.000
Mutual assistance (five-point Likert scale)	3.702	0.909	1.000	5.000
Trust in neighbors (five-point Likert scale)	3.909	0.774	1.000	5.000
Comparison with relatives (five-point Likert scale)	2.686	0.746	1.000	5.000
Comparison with neighbors (five-point Likert scale)	2.763	0.655	1.000	5.000
Gender (1 = male; 0 = female)	0.517	0.500	0.000	1.000
Age	50.689	12.553	18.000	87.000
Education level (never went to school = 0, primary school = 1, junior high school = 2, senior high school/technical school/technical secondary school = 3, college and above = 4)	1.515	0.985	0.000	4.000
Married (1 = yes; 0 = otherwise)	0.898	0.303	0.000	1.000
Religious belief (1 = yes; 0 = otherwise)	0.108	0.310	0.000	1.000
Insurance status (1 = yes; 0 = otherwise)	0.942	0.233	0.000	1.000
Exercise habit (1 = yes; 0 = otherwise)	0.224	0.417	0.000	1.000
Household income (logged annual house income)	9.968	2.077	0.000	13.122
Number of brothers and sisters	3.526	1.991	0.000	8.000

**Table 2 ijerph-20-02213-t002:** Baseline result for the impact of migration experience on rural residents’ mental health.

	MH			
	(1)	(2)	(3)	(4)
ME	0.121 ***	0.128 ***	0.127 ***	0.122 ***
	(0.030)	(0.030)	(0.030)	(0.030)
Gender	−0.208 ***	−0.167 ***	−0.171 ***	−0.170 ***
	(0.021)	(0.022)	(0.022)	(0.022)
Age	0.020 ***	0.022 ***	0.021 ***	0.018 ***
	(0.006)	(0.006)	(0.006)	(0.007)
Age-squared	−0.000 **	−0.000 ***	−0.000 ***	−0.000 ***
	(0.000)	(0.000)	(0.000)	(0.000)
Education		−0.123 ***	−0.115 ***	−0.100 ***
		(0.014)	(0.014)	(0.015)
Married		−0.199 ***	−0.200 ***	−0.177 ***
		(0.044)	(0.044)	(0.044)
Religious belief		0.051	0.045	0.038
		(0.044)	(0.044)	(0.044)
Insurance status			−0.005	0.007
			(0.056)	(0.056)
Exercise habit			−0.103 ***	−0.088 ***
			(0.030)	(0.030)
Household income				−0.050 ***
				(0.006)
Number of brothers and sisters				0.021 ***
				(0.007)
Observations	7790	7790	7790	7790

Note: ** *p* < 0.05, *** *p* < 0.01.

**Table 3 ijerph-20-02213-t003:** Robustness checks: alternative measures of mental health.

	MH (a)	MH (b)	Happiness	Satisfaction
	(1)	(2)	(3)	(4)
ME	0.124 ***	0.211 ***	−0.090 ***	−0.124 ***
	(0.047)	(0.038)	(0.032)	(0.032)
Control variables	Yes	Yes	Yes	Yes
Observations	7790	7790	7790	7790

Note: *** *p* < 0.01.

**Table 4 ijerph-20-02213-t004:** Robustness checks: alternative specifications.

Coefficient	ATT	Standard Error	Observations
*Panel A: adjust for village-level clustering*			
0.122 ***		0.042	7790
*Panel B: ologit estimates*			
0.212 ***		0.050	7790
*Panel C: IV-oprobit estimates*			
0.795 ***		0.064	7790
*Panel D: Heckman two-step methods*			
0.100 ***		0.030	7790
*Panel E: use the PSM to estimate the ATT*			
	0.676 *	0.355	7790

Note: * *p* < 0.1, *** *p* < 0.01.

**Table 5 ijerph-20-02213-t005:** Mechanism analysis: social comparison.

	Comparison with Relatives	MH	Comparison with Neighbors	MH
	(1)	(2)	(3)	(4)
ME	−0.148 ***	0.096 ***	−0.106 ***	0.104 ***
	(0.032)	(0.030)	(0.033)	(0.030)
Comparison with relatives		−0.294 ***		
		(0.018)		
Comparison with neighbors				−0.330 ***
				(0.020)
Control variables	Yes	Yes	Yes	Yes
Observations	7790	7790	7790	7790

Note: *** *p* < 0.01.

**Table 6 ijerph-20-02213-t006:** Mechanism analysis: social support.

	Mutual Assistance	MH	Trust in Neighbors	MH
	(1)	(2)	(3)	(4)
ME	−0.145 ***	0.108 ***	−0.174 ***	0.102 ***
	(0.032)	(0.030)	(0.033)	(0.030)
Mutual assistance		−0.110 ***		
		(0.014)		
Trust in neighbors				−0.159 ***
				(0.017)
Control variables	Yes	Yes	Yes	Yes
Observations	7790	7790	7790	7790

Note: *** *p* < 0.01.

**Table 7 ijerph-20-02213-t007:** Heterogeneous Effects: different types of returning.

	MH					
	(1)	(2)	(3)	(4)	(5)	(6)
Long-term returning	0.086 ***		0.102 ***			
	(0.031)		(0.032)			
Short-term returning		0.169 ***	0.195 ***			
		(0.054)	(0.055)			
Voluntary returning				−0.003		0.011
				(0.038)		(0.038)
Passive returning					0.209 ***	0.210 ***
					(0.047)	(0.048)
Control variables	Yes	Yes	Yes	Yes	Yes	Yes
Observations	7790	7790	7790	7790	7790	7790

Note: *** *p* < 0.01.

**Table 8 ijerph-20-02213-t008:** Heterogeneous Effects: age and village appearance.

	Age		Village Appearance	
	Old Aged (1)	Young and Middle-Aged (2)	Good (3)	Ordinary (4)
ME	0.054	0.146 ***	0.054	0.135 ***
	(0.057)	(0.035)	(0.065)	(0.034)
Control variables	Yes	Yes	Yes	Yes
Observations	2867	4923	2028	5762

Note: *** *p* < 0.01.

## Data Availability

The data were released to the researchers without access to any personal data. Data access link: http://css.sysu.edu.cn/ (accessed on 28 December 2022).

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
