# Peer review of "The Impact of Migration Experience on Rural Residents’ Mental Health: Evidence from Rural China"

_ijerph, 2023, doi:10.3390/ijerph20032213_

Round 1

Reviewer 1 Report

         This is an interesting and meaningful study. Using the 2018 CLDS survey data, the authors systematically analyzed the impact of labor migration experience on their mental health. Here are some suggestions for your reference:

         (1) The measurement problem of core variables. The core variable that the author is concerned about is Rural-Urban Migration Experience. However, the author mentions the return of labor force in many places in the text. In fact, these two concepts are different and the author needs to clearly define them. At the same time, the labor force may return for many times during the years of migrant work experience, and some labor force may change more than one job in a year. Different mobility experiences may have an impact on their mental health, but the author does not seem to test these problems.

         (2) The mechanism of action between core variables. The industry of labor migration, location, time, place of migration and environment of the place of migration are all important factors affecting their mental health. However, the author seems to have not considered these issues. To be honest, although the author has now concluded that the experience of rural-urban migration will affect the mental health of the labor force, I think the author has not grasped the key of this, at least the theoretical analysis has not been sorted out, and the role path needs to be further sorted out.

         (3) The part of heterogeneity analysis can be further strengthened. I don't know whether the scale of the author's analysis is individual or family? A family may have more than one migrant worker, how does the author deal with this problem? If the individual is taken as the object of analysis, then whether family members, especially spouses, move with them is also an important factor affecting their mental health, the author can make a more in-depth discussion on these issues.

Reviewer 2 Report

The paper is well organized, while, I would have to say, it is an interesting topic and a good research topic in rural China, However, the author hasn’t provided a very rigorous research analysis framework and research hypotheses. Overall, the paper is writing well, but need to improved as my comments below, based on these, I can consider accept the paper after the modification and revise, my specific comments are in the bellowing:

(1) The abstract hasn’t been well organized, you should simply put forward you core research question, and then your research method and theory used in the paper, and then your research findings and the possible conclusion and policy recommendation for the practice. In the current status, the abstract is too complex, which need to simplify. In addition, there hasn’t any policy recommendation for the rural villages and communities.

(2) I found the authors haven’t organized well the literature review, when you organized your literature review, you cannot list the opinions of the researchers, which researches that the authors have done, you should try to make a summarization, regarding on one aspect, a lot of authors have focus on this, and for another aspect, some other authors have focus on that. It need summarization and then evaluation, then you can find the research gap, and confirm that the research questions that you focus on are important.

(3) The authors have presented the marginal contributions of the paper, this is good, however, I think you only need to present what’s the creatives that you have done than other previous scholars, while you no need to presented what’s your findings in this section, it will need move to the results and mechanism analysis.

(4)  The major problems lie in that the paper has not a very rigorous theoretical framework. I Find that the paper has talked much about the mental health, talked much about the migration experience, which you have not put forward a theoretical framework to tell the readers, what is the influence mechanism, what role that social comparative and social support will act in this influence route, act as a positive mediating role or a negative mediating role. if these are your scientific research problem. You should try to construct a good theoretical framework, and you should better construct a theoretical framework diagram, which can make the route clearer for the readers.

(5) The paper has used sufficient econometric models to solve the estimation problems and have do the robustness check and also some endogeneity problems, which is deserves to be admired, while the control variable, if there are some missing, you need to take them into consideration, and also explain clearly why you choose these control variables.

(6) In the mechanism analysis section, you find that both the social support and social comparison have a partial mediating role in the relationship between migrant worker experience and returning migrants workers’ mental health, I think you can calculate the influence coefficients and reported them in your results and in the abstract. The authors have given some good policy recommendations, these policies are specific and can be available, which need to admired, what I am concern, if the policy recommendation can better link to the regression results and mechanism analysis, it will be better and considerable.

Round 2

Reviewer 1 Report

I have no other comments, thank you.